# Unravelling the Complexity of the +33 C>G [HBB:c.-18C>G] Variant in Beta Thalassemia

**DOI:** 10.3390/biomedicines12020296

**Published:** 2024-01-27

**Authors:** Coralea Stephanou, Miranda Petrou, Petros Kountouris, Christiana Makariou, Soteroula Christou, Michael Hadjigavriel, Marina Kleanthous, Thessalia Papasavva

**Affiliations:** 1Molecular Genetics Thalassemia Department, The Cyprus Institute of Neurology and Genetics, Nicosia 2371, Cyprus; 2Thalassemia Clinic Nicosia, Archbishop Makarios III Hospital, Nicosia 2012, Cyprus; 3Thalassemia Clinic Limassol, Limassol General Hospital, Limassol 4131, Cyprus

**Keywords:** β-thalassemia intermedia, HBB, 5′UTR, silent variant, genotype/phenotype correlation

## Abstract

The +33 C>G variant [NM_000518.5(HBB):c.-18C>G] in the 5′ untranslated region (UTR) of the β-globin gene is described in the literature as both mild and silent, while it causes a phenotype of thalassemia intermedia in the presence of a severe β-thalassemia allele. Despite its potential clinical significance, the determination of its pathogenicity according to established standards requires a greater number of published cases and co-segregation evidence than what is currently available. The present study provides an extensive phenotypic characterization of +33 C>G using 26 heterozygous and 11 compound heterozygous novel cases detected in Cyprus and employs computational predictors (CADD, RegulomeDB) to better understand its impact on clinical severity. Genotype identification of globin gene variants, including α- and δ-thalassemia determinants, and rs7482144 (XmnI) was carried out using Sanger sequencing, gap-PCR, and restriction enzyme digestion methods. The heterozygous state of +33 C>G had a silent phenotype without apparent microcytosis or hypochromia, while compound heterozygosity with a β+ or β0 allele had a spectrum of clinical phenotypes. Awareness of the +33 C>G is required across Mediterranean populations where β-thalassemia is frequent, particularly in Cyprus, with significant relevance in population screening and fetal diagnostic applications.

## 1. Introduction

β-thalassemia results from quantitative defects in the β-subunit of adult hemoglobin (HbA, α_2_β_2_). Thus far, over 400 different mutant alleles have been reported that affect different levels of β-globin (*HBB*) gene regulation and expression [1]. They are generally classified as β^0^ when no chains are produced, β^+^ when β-globin production is severely reduced, and β^++^ when β-globin production is mildly reduced [2]. They have a distinct hematological phenotype in the heterozygote, essentially characterized by a variable reduction in mean cell hemoglobin (MCH) and mean cell volume (MCV) with raised HbA_2_ levels. In some cases, β alleles are so mild that they are phenotypically silent with normal hematological parameters. Interactions of β alleles produce variable phenotypes ranging from a transfusion-dependent to an intermediate-to-mild form of anemia [3]. The clinical severity of the condition may be modified by the co-inheritance of ameliorating genetic factors such as α-thalassemia and genetic determinants related to fetal hemoglobin (HbF) [4].

The presence of mild β alleles usually leads to intermediate forms of thalassemia during adulthood or in later stages of life. They manifest a condition of intermediate severity in the homozygous state, whereas interactions with severe alleles confer a spectrum of severity ranging from mild to severe. The mildest of the β alleles, also called silent, exert a hematological phenotype typical of heterozygous β-thalassemia when present in the homozygous state and a mild form of thalassemia intermedia in the compound heterozygous state with severe alleles [5,6]. A few mild and even less silent β alleles have been identified thus far, mainly affecting *HBB* transcription, mRNA processing, and mRNA translation [2,7], as shown in Table 1. The identification and characterization of mild β alleles have important implications for fetal diagnostic and treatment services. This is particularly important due to recent findings linking milder genotypes to poorer survival [8], while the accurate pathogenicity assertion of the mild β thalassemia variants remains a priority for clinical genome interpretation.

The present study reports novel case-level data on β +33 C>G [NM_000518.5(HBB):c.-18C>G] single nucleotide substitution located in the 5′ untranslated region (UTR) of *HBB*. This variant is characterized by a few studies primarily conducted in Mediterranean populations to cause minimally reduced or normal red cell indices in the heterozygous state. However, when combined with severe β^+^ or β^0^ alleles, it manifests a thalassemia intermedia phenotype [9,10]. Variant-to-phenotype associations have been described so far for four heterozygotes and six compound heterozygotes (available in IthaPhen [11]). Case-level and segregation data are important in the evaluation of variant pathogenicity using the American College of Medical Genetics and Genomics and the Association for Molecular Pathology (ACMG/AMP) best-practice guidelines [12,13] while insufficient numbers of independent cases can contribute to the classification of variants as variants of uncertain significance (VUS). The present study enriches the current knowledge on the genotype–phenotype landscape of mild β-thalassemia through the description of 26 heterozygous and 11 compound heterozygous novel β +33 C>G cases from the Cypriot population. Moreover, the study employs computational predictions to assess the variant’s functional consequence with the aim of generating a more precise understanding of its impact on clinical severity.

**Table 1 biomedicines-12-00296-t001:** Mild and silent β alleles causing β-thalassemia. Reproduced from Thein 2013 [2] and Cao and Galanello [14].

Variant Effect		Location (NG_)	Mild β	Silent β	IthaGenes ID
**Transcription**	promoter	c.-151C>T		-101 C>T	IthaID: 3
		c.-151C>G		-101 C>G	IthaID: 4
		c.-142C>T		-92 C>T	IthaID: 6
		c.-140C>T	-90 C>T		IthaID: 7
		c.-138C>T	-88 C>T		IthaID: 8
		c.-138C>A	-88 C>A		IthaID: 9
		c.-137C>G	-87 C>G		IthaID: 10
		c.-137C>T	-87 C>T		IthaID: 11
		c.-137C>A	-87 C>A		IthaID: 12
		c.-136C>A	-86 C>A		IthaID: 14
		c.-136C>T	-86 C>T		IthaID: 4028
		c.-136C>G	-86 C>G		IthaID: 13
		c.-123A>T	-73 A>T		IthaID: 15
		c.-81A>G	-31 A>G		IthaID: 20
		c.-80T>A	-30 T>A		IthaID: 22
		c.-79A>G	-29 A>G		IthaID: 25
	5’UTR	c.-50A>C		CAP +1 A>C	IthaID: 34
		c.-43C>T		CAP +8 C>T	IthaID: 35
		c.-41delT	CAP +10 (-T)	CAP +10 (-T)	IthaID: 36
		c.-29G>A	CAP +22 G>A		IthaID: 38
		c.-18C>G	CAP +33 C>G	CAP +33 C>G	IthaID: 39
**RNA processing**	Consensus splice site	c.92+6T>C	IVS I-6 T>C		IthaID: 111
		c.316-7C>A		IVS II-844 C>A	IthaID: 219
		c.316-7C>G		IVS II-844 C>G	IthaID: 220
	Cryptic splice site	c.59A>G	CD19 AAC>AGC Hb Malay		IthaID: 79
		c.75T>A	CD 24 GGT>GGA		IthaID: 86
		c.82G>T		CD 27 GCC>TCC Hb Knossos	IthaID: 91
	Poly A site	c.*110T>C	AATAAA>AACAAA		IthaID: 272
		c.*111A>G	AATAAA>AATGAA		IthaID: 274
		c.*112A>G	AATAAA>AATAGA		IthaID: 275
		c.*113A>G	AATAAA>AATAAG	AATAAA>AATAAG	IthaID: 276
		c.*108A>C		AATAAA>CATAAA	IthaID: 270
	3’UTR	c.*6C>G		Term CD +6 C>G (CAP +1480)	IthaID: 267
		c.*93_*105del		Term CD +90, del 13 bp (CAP +1567 to +1579)	IthaID: 269
		c.*47C>G	Term CD +47 C>G		IthaID: 268

## 2. Materials and Methods

### 2.1. Ethics Statement and Study Participants

The study was conducted in the Molecular Genetics Thalassemia Department of the Cyprus Institute of Neurology and Genetics (CING), in collaboration with the Thalassemia Clinics in Nicosia and Limassol. Participants provided their informed consent for the analysis. Samples were collected as part of routine patient care and data were extracted from the clinical records of participants. Molecular analyses were carried out in the CING laboratory, adhering to the CYS EN ISO 15189:2012 accreditation standard, in response to participant requests. The results underwent thorough evaluation and confirmation by the senior head of the laboratory before being sent to the ordering physician. This study is a report of the clinical experience of the assay concealing any description or personal information of the individual case. All data were de-identified to protect the privacy of participants and submitted to IthaPhen (https://www.ithanet.eu/db/ithaphen) [11], accessed on 18 September 2023.

### 2.2. Hematological Analyses

Hematological studies were performed using routine methods. Hemoglobin (Hb) analysis, including the separation and quantification of Hb subtypes, was performed by cation exchange high-performance liquid chromatography (HPLC) (VARIANT™; Bio-Rad Laboratories, Hercules, CA, USA).

### 2.3. Molecular Analyses

Genomic DNA was isolated from peripheral blood using the Puregene Blood Core Kit C (Qiagen Sciences, Germantown, MD, USA). Samples were investigated for β-globin variants [15] and, where indicated through abnormal hematological indices, δ-globin variants [15] by Sanger sequencing using the BigDye Terminator v3.1 Cycle Sequencing Kit (Applied Biosystems, Woburn, MA, USA). Samples were tested for 8 common α-globin deletions and/or point mutations by gap-PCR [15] and restriction enzyme digestion (RED), namely -α3.7, ααα, --MEDI, -α20.5, α5ntα, αAgrα, αPolyA2α, αPolyA1α, -α4.2, αIcariaα, and αcd108(-C), where α-thalassemia was indicated (reduced MCV and/or MCH but normal/borderline-raised HbA_2_). Analysis of the XmnI C>T change [NG_000007.3(HBG2):g.42677C>T] at position -158 of the HBG2 gene was determined by PCR-restriction fragment length polymorphism (PCR-RFLP) using primers described previously [16].

### 2.4. Bioinformatics Analyses

The Combined Annotation Dependent Depletion (CADD) [17,18,19] tool was used to predict the deleteriousness of the 5′UTR β +33 C>G variant. The PHRED CADD score for this variant was accessed from the Ensembl Variant Effect Predictor (VEP) [20] tool and was compared to thresholds for pathogenic and benign predictions for variants in hemoglobinopathy-related globin genes as defined in Tamana et al. [21]. The RegulomeDB v2.2 [22] tool was used to annotate regulatory information on this variant and facilitate the identification of its potential causal link to disease. RegulomeDB v2.2 scores range from 1 to 6, with lower scores indicating a higher probability of a functional variant.

## 3. Results

### 3.1. Case-Level and Segregation Data

This study presents for the first time genotype–phenotype data on β +33 C>G [NM_000518.5(HBB):c.-18C>G] in 26 heterozygous and 11 compound heterozygous individuals of Cypriot ethnicity. Table 2 summarizes the hematological and clinical phenotypic characteristics of the study population. The relatedness and structure of the six families included in this study are shown in Appendix A. The variant was identified during the molecular investigation of samples with abnormal hematological indices and as part of specialized diagnostic services following a request for prenatal or pre-implantation genetic diagnosis.

### 3.2. Heterozygotes for β +33 C>G

Phenotypic descriptions are presented for 26 heterozygous individuals. Heterozygous β-thalassemia is commonly detected by the presence of microcytosis (MCV < 79 fL), hypochromia (MCH < 27 pg), and an increased level of HbA_2_ (≥3.5%). The β^++^ alleles have higher MCV and MCH values than β^+^ and β^0^ alleles although lower than normal, and a borderline-normal to slightly increased HbA_2_. The silent alleles of the β^++^ group usually have normal RBC indices and HbA_2_, although borderline-raised HbA_2_ and/or slightly reduced MCV and MCH values have been observed [23]. Among the study sample of 26 heterozygous cases, the majority exhibited normal MCV (85.19 ± 5.39 (mean ± SD), 85.4 (69.9–93.5) (median (min–max))) and MCH (28.12 ± 2.04 (mean ± SD), 28.3 (22.7–31.3) (median (min–max))) values with normal/borderline-raised HbA_2_ levels (3.02 ± 0.27 (mean ± SD), 3 (2.4–3.6) (median (min–max))), as shown in Table 2. Atypical cases with reduced MCV and/or MCH but normal/borderline-raised HbA_2_ (cases 3, 6, 7, 9, and 12), as shown in Table 3, were further studied for co-transmitted α-thalassemia or double heterozygosity with δ-thalassemia. Screening for α-thalassemia determinants revealed the presence of the -α^3.7^ deletion [NG_000006.1:g.34247_38050del] in a subset of samples, with the inactivation of two α globin genes (-α^3.7^/-α^3.7^) having a greater impact on RBC indices compared to the loss of one α globin gene (-α^3.7^/αα). No *HBD* variants were detected in any of the samples. All cases that raised suspicion for the presence of a β^++^ allele (borderline MCV, MCH, and/or HbA_2_) were unequivocally diagnosed with heterozygous β +33 C>G through direct sequencing analysis. Upon diagnosis, all heterozygous individuals exhibited a normal clinical phenotype.

### 3.3. Compound Heterozygotes for β +33 C>G with Other β-thalassemia Alleles

Phenotypic data are presented for 11 compound heterozygotes with β alleles that are frequent in the Cypriot population [15], namely β^+^ IVS I-110 G>A [HBB:c.93-21G>A] (6 cases), β^0^ CD 39 C>T [HBB:c.118C>T] (4 cases), and β^0^ IVS I-1 G>A [HBB:c.92+1G>A] (1 case), as shown in Table 4. In none of the samples analyzed was homozygosity for the β +33 C>G variation detected. Among the study sample, six individuals required medical intervention, which eventually led to a diagnosis of β-thalassemia (cases 1–6, Table 4), while the remaining samples were detected during family studies and routine screening. Hematological indices were reported at the time of diagnosis (pre-transfusion) for ten patients, revealing a similar pattern of microcytic, hypochromic anemia (MCV 60.28 ± 3.59 (mean ± SD), 61.05 (54.5–65.2) (median (min–max)); MCH 19.22 ± 1.33 (mean ± SD), 19.4 (17.2–21.3) (median (min–max); Hb (female) 8.86 ± 0.56 (mean ± SD), 8.7 (8–9.6) (median (min–max)); Hb (male) 10.13 ± 3.11 (mean ± SD), 8.7 (8–13.7) (median (min–max))), as shown in Table 1. All patients had a normal alpha genotype except for case 7 (with β^+^ IVS I-110 G>A), which had heterozygous α^+^ thalassemia (αα/-α^3.7^) and had also tested positive for the Xmn*I* polymorphism. Owing to the co-inheritance of these ameliorating factors affecting disease severity, case 7 presented with a near-normal Hb level of 13.7 g/dL and mild morphologic abnormalities of blood cells on film, with no evident clinical manifestations. Another patient (case 9, with β^+^ IVS I-110 G>A) was found to carry the Xmn*I* polymorphism, which likely contributed to an increase in HbF at 8.2% but without profound improvements in the patient’s hematological phenotype. This patient had no history of blood transfusion, while clinical data were unavailable. Three patients had transfusion-dependent thalassemia (TDT) of whom two patients required regular blood transfusions in adulthood after the age of 30 (cases 3 and 4, both with β^0^ CD 39 C>T), while the third patient began transfusion therapy at the young age of three (case 6, with β^+^ IVS I-110 G>A). Of these, only case 3 presented with clinical symptoms, namely splenomegaly and facial bone deformities. Two other patients (cases 1 and 8) required occasional blood transfusions following medical procedures but were otherwise clinically normal. Two out of three patients with non-transfusion-dependent thalassemia (NTDT) presented with splenomegaly (case 5, with β^+^ IVS I-110 G>A) and a barely palpable spleen (case 11, with β^0^ IVS I-1 G>A), while the third patient only had an abnormal hematological phenotype (case 2, with β^0^ CD 39 C>T). Overall, case 6 appears to have the most severe phenotype among all diagnosed patients due to an early-on blood transfusion requirement; the presence of an alternate genetic determinant of disease severity is probable, warranting further investigation.

### 3.4. A Use Case for Comprehensive Genetic Analysis

Clinical diagnostic laboratories, including the CING laboratory, are increasingly adopting genetic testing protocols that rely on next-generation sequencing (NGS) technologies. These advanced methods enable the comprehensive detection of various genetic variations, facilitating the diagnosis of hemoglobinopathies and other inherited diseases. Figure 1 illustrates a family that underwent β-thalassemia prenatal diagnosis at the CING. The mother is a carrier of the severe IVS I-110 G>A variation while the father (case 7, Table 4) was initially diagnosed as a carrier of the IVS I-110 G>A variation also through targeted molecular methods. Sanger sequencing of the *HBB* performed on the fetal sample for prenatal diagnosis purposes detected the β +33 C>G variation in the heterozygous state. Repeat parental testing identified β +33 C>G in the father in compound heterozygosity with IVSI-110 G>A. IVSI-110 G>A is a severe β^+^ allele, inducing abnormal splicing and leading to a reduced steady-state HBB protein level of only 10% in affected individuals [24]. This case of fetal diagnostics demonstrates the silent nature of β +33 C>G in the presence of a severe β allele in the hematological phenotype of the father and highlights the benefit of adopting more comprehensive approaches, such as NGS, for holistic thalassemia genetic profiling.

### 3.5. Computational Data

The computational predictor CADD annotated β +33 C>G (rs34135787) as pathogenic with a PHRED CADD score of 18 (threshold > 12) [21]. RegulomeDB identified this variant as having a potentially functional consequence on *HBB* by exploring various sources of data, including chromatin states, epigenetic marks, motif instances, transcription factor (TF) binding, and expression quantitative trait loci (eQTLs) annotations, as shown in Figure 2. β +33 C>G (rs34135787) is located 33 bases downstream of the transcriptional start site of *HBB*. RegulomeDB showed hits to several Dnase-seq peaks (open chromatin regions) in various cell types with significant activity in hematopoietic multipotent progenitor cells. In this cell type, this region (rs34135787-*HBB*) was mapped with an active enhancer state and shown to contain TF binding sites for GATA1 (GATA binding protein 1), POLR2A (RNA Polymerase II Subunit A), and CEBPA (CCAAT enhancer binding protein alpha). Furthermore, RegulomeDB TF motif evidence suggested that allele G would disrupt the binding of RFX2 (DNA-binding protein RFX2) in liver-related biosamples and RUNX3 (RUNX3 antisense RNA 1) in 26 different cells and tissues. RegulomeDB scored rs34135787 as a category 2 variant based on evidence of binding through ChIP-seq and DNase data albeit in the absence of eQTL information, indicating that this variant is most likely to impact gene regulation.

## 4. Discussion

β +33 C>G (rs34135787) is a non-coding 5′UTR variant in *HBB* characterized in the literature as both a mild and silent β thalassemia allele [2,9,10,14], while its pathogenicity is not fully determined. Published case reports on β +33 C>G are scarce, while this variant is absent from reference population databases, indicating that this is possibly a rare variant. It was previously detected in individuals of Spanish [10], Turkish [25], and Greek Cypriot [9] origin, while a recent epidemiological study revealed that β +33 C>G has a carrier frequency of 0.086% (2/2335 alleles) in Cyprus [15]. This evidence shows that β +33 C>G is most likely to be encountered in people of Mediterranean heritage, particularly the population of Cyprus, having significant relevance in population screening and fetal diagnostic applications.

The present study provides the most extensive phenotypic characterization of β +33 C>G using 26 heterozygous and 11 compound heterozygous cases detected in Cyprus as part of routine screening and specialized molecular diagnostic procedures. The vast majority of heterozygous samples had normal RBC indices and a normal/borderline-raised HbA_2_ level. A few atypical cases were analyzed for the presence of α-thalassemia and δ-thalassemia determinants. Only the deletional -α^3.7^ variant was detected, which is the most common α-thalassemia allele in Cyprus with a relative frequency of 72.8% of all α-globin variations [15]. Overall, the heterozygous state of β +33 C>G exhibited a mild effect on the hematological phenotype. On the other hand, co-inheritance of β +33 C>G with a severe β allele produced hematological changes and a spectrum of clinical phenotypes among the affected individuals, while anemia was exacerbated under certain conditions, such as pregnancy. Co-inheritance with a β^+^ allele (IVS I-110 G>A) demonstrated a milder variable impact on the observed phenotype compared to the co-inheritance with a β^0^ allele (CD 39 C>T, IVS I-1 G>A). Overall, the co-inheritance with a severe β^0^ or β^+^ allele produced a clinical phenotype of thalassemia intermedia that worsened with age. The phenotypic manifestation observed in cases 6 and 7 (Table 4) may have been influenced by additional hereditary factors, indicating the need for further investigation. Overall, the phenotype of compound heterozygotes varied from clinically silent to mild transfusion-dependent thalassemia, indicating that a larger number of patients will be needed to delineate the pathogenicity of the variant and facilitate accurate prediction of the phenotype.

The mild and silent β alleles are predominantly identified incidentally during routine diagnostic screening or during investigating parental inheritance patterns. They are predominantly found in the non-coding sequences of *HBB* and can impact β-globin expression mainly by affecting *HBB* gene transcription (promoter, 5′UTR) and mRNA processing (consensus and cryptic splice sites, poly A signal, 3′UTR) [2,14]. In addition, the 5′UTR sequences may contain structural motifs and upstream open reading frames that regulate mRNA translation [26]. In vitro cell-based assays investigating 5′UTR β alleles previously associated with thalassemia intermedia showed that the β +33 C>G allele acts at the level of transcription by reducing *HBB* mRNA output without affecting its stability [27,28]. Its genomic location also suggests a role in the regulation of *HBB* transcription. The +33 position lies downstream of the *HBB* transcriptional start site within a region that contains the downstream core element (DCE). The C>G change at this position disrupts DCE and, in turn, the binding of transcription factor II D (TFIID) that aids in the recruitment of RNA polymerase II (RNAPII) to the promoter [29]. These findings are in line with RegulomeDB v2.2 functional annotation of β +33 C>G. The predicted binding of GATA1 (OMIM 305371, regulates RNAPII recruitment to the HBB promoter and BLCR) and POLR2 (OMIM 180660, encodes the large subunit of RNAPII) and the active enhancer state of this region (rs34135787-*HBB*) in blood-derived cells showed that β +33 C>G is likely to function by disputing RNAPII-mediated *HBB* expression. Collectively, these data demonstrated that the major mechanism by which β +33 C>G reduces the production of *HBB* mRNA is at the level of transcription.

In conclusion, β +33 C>G is a mild β allele associated with a thalassemia intermedia phenotype, exhibiting cosegregation with disease across multiple affected families and is supported by pathogenic predictions by computational methods and functional analyses. It is expected that there are many more cases of β +33 C>G that remain undiagnosed since the disease usually manifests at an older age, with or without clinical intervention, or is exacerbated during particular conditions such as infective diseases and pregnancy. Diagnostic screening for hemoglobinopathies traditionally uses molecular approaches to search for the most frequent genetic variations in the population from which the patient originates, resulting in mild and silent variations almost often remaining unreported. A showcase of fetal diagnostics highlights the benefit of implementing NGS to enable comprehensive and accurate detection of genetic variation. Although there is currently no indication for prenatal diagnosis in mild or even silent β-thalassemia, genotype information can equip prospective parents with the knowledge that their offspring may require medical care, such as occasional transfusions during the course of their lives.

The present study describes for the first time the phenotype of β +33 C>G in the Cypriot population and provides the largest pool of samples for comprehensive genotype–phenotype analysis to better predict the clinical effect of this variant. The sharing of case observations and phenotypic information from laboratories to public domains serves as an invaluable resource for clinical variant interpretation and resolving variants with conflicting interpretations. It is also valuable for the study of rare diseases, such as hemoglobinopathies and other rare anemias that often have limited sample sizes and fragmented data. The genotype–phenotype data featured in this paper is openly available on the IthaPhen database [11]. Despite detailing the minimum standards in screening and diagnosis in this study, it is essential to acknowledge the importance of a more thorough understanding. This can be achieved through the implementation of larger and more detailed future studies, which are necessary to fully elucidate the genotype–phenotype correlations within this cohort. Furthermore, the study highlights the importance of early detection since the co-inheritance of a mild β allele, as in the case of β +33 C>G, with a severe β allele may lead to a clinical phenotype with important implications on the life of the affected individual. Genotype information can play a crucial role in enabling affected individuals to proactively seek therapeutic management for their condition and receive proper care at an early stage, ultimately contributing to an improved quality of life.

## Figures and Tables

**Figure 1 biomedicines-12-00296-f001:**
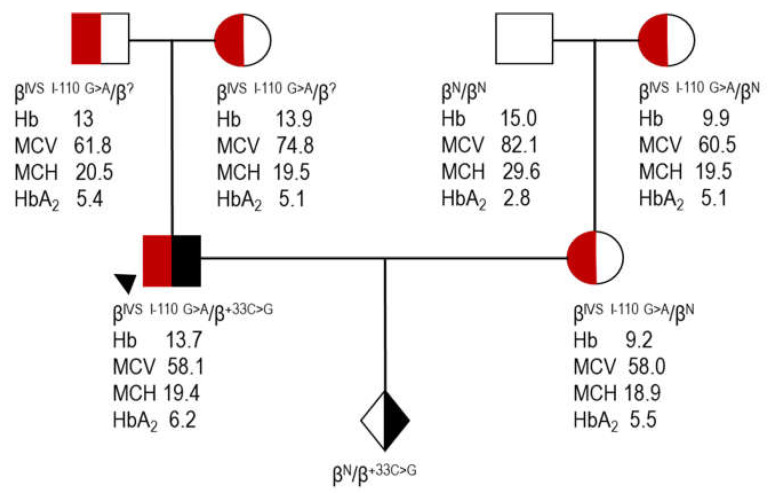
A pedigree showing *HBB* carriers and affected family members with IVS I-110 G>A and +33 C>G. 
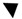
 proband (case 7, Table 4); Hb, hemoglobin; MCV, mean corpuscular volume; MCH, mean corpuscular hemoglobin; HbA_2_, A_2_ hemoglobin; ?, unknown β allele as a result of direct detection methods utilized for the identification of the most common β variants.

**Figure 2 biomedicines-12-00296-f002:**
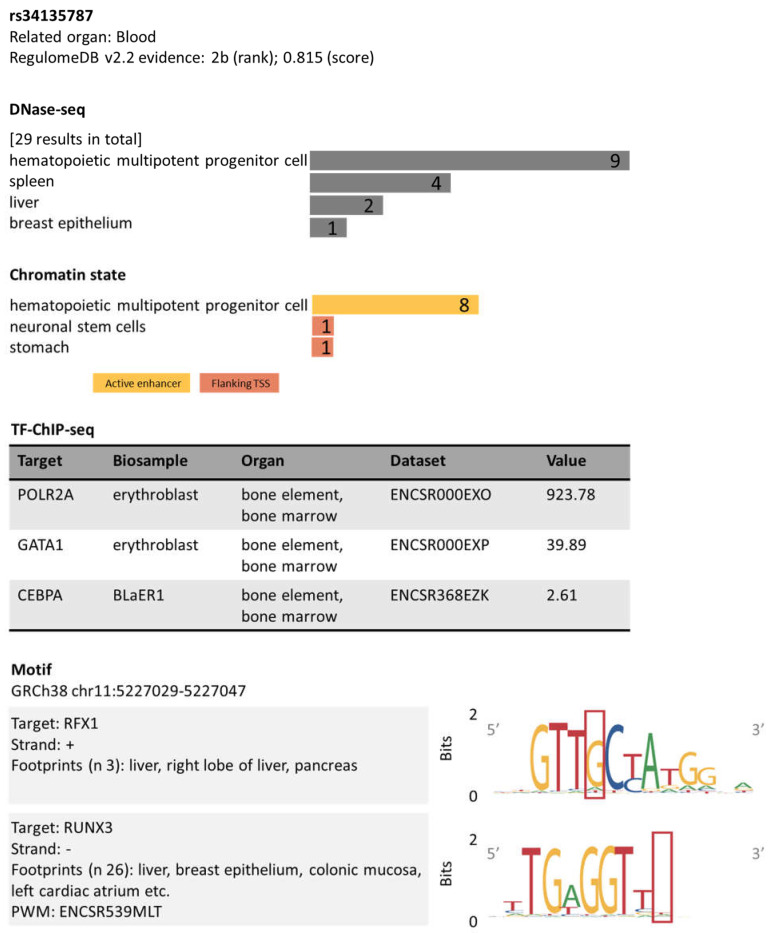
RegulomeDB v2.2 analysis for rs34135787 [Accessed: 30 May 2023]. RegulomeDB scores the functionality of rs34135787 based upon experimental data derived from large published datasets covering various tissues and cell lines, such as its presence in DNase hypersensitive regions (DNase-seq), promoters or enhancers (chromatin state), and sequences affecting the binding of transcription factors (TF-ChIP-seq) and DNA motifs. The top hits in each category are shown.

**Table 2 biomedicines-12-00296-t002:** Characteristics of the study sample.

	Heterozygotes			Compound Heterozygotes		
Variable	*n*	Mean ± SD	Median (min–max)	*n*	Mean ± SD	Median (min–max)
Sex at birth						
Female	16			7		
Male	10			4		
Age (years)						
Female	16	37.4 ± 17.38	30 (16–66)	7	37.43 ± 17.9	34 (15–61)
Male	10	32.2 ± 14.7	29 (17–70)	4	25.5 ± 19.33	23.5 (4–51)
Hb (g/dL)						
Female	12	13.14 ± 0.96	13.4 (11.6–14.3)	7	8.86 ± 0.56	8.7 (8–9.6)
Male	6	15.47 ± 1.17	15.6 (13.7–17.2)	3	10.13 ± 3.11	8.7 (8–13.7)
RBC (×10^6^)						
Female	12	4.72 ± 0.3	4.73 (4.23–5.35)	7	4.62 ± 0.48	4.56 (3.99–5.24)
Male	6	5.38 ± 0.43	5.28 (4.98–6.22)	3	5.31 ± 1.54	4.64 (4.21–7.07)
PCV (%)						
Female	12	39.9 ± 1.99	40.05 (36.4–42.4)	7	28.01 ± 2.49	26.8 (25.2–31.9)
Male	6	46.02 ± 2.95	46.55 (41.7–49.9)	3	31.23 ± 8.6	27.3 (25.3–41.1)
MCV (fL)	26	85.19 ± 5.39	85.4 (69.9–93.5)	10	60.28 ± 3.59	61.05 (54.5–65.2)
MCH (pg)	26	28.12 ± 2.04	28.3 (22.7–31.3)	10	19.22 ± 1.33	19.4 (17.2–21.3)
MCHC (g/dL)	18	33.05 ± 0.99	33.05 (30.5–34.7)	10	31.87 ± 1.0	31.85 (29.8–33.3)
HbA_2_ (%)	26	3.02 ± 0.27	3 (2.4–3.6)	9	5.84 ± 0.37	5.8 (5.2–6.4)
RDW (%)	10	13.89 ± 1.8	13.55 (11.9–18.7)			
HbF (%)				7	3.97 ± 2.71	3.1 (1.5–8.2)
Transfusion phenotype						
NTDT				6		
TDT				3		
Bone deformities						
Present				1		
Absent				8		
Splenomegaly						
Present				3		
Absent				6		
Hepatomegaly						
Present				0		
Absent				8		
Splenectomy						
Yes				0		
No				9		

Hb, hemoglobin; HbA_2_, A_2_ hemoglobin; HbF, F hemoglobin; MCH, mean corpuscular hemoglobin; MCV, mean corpuscular volume; MCHC, mean corpuscular hemoglobin concentration; NTDT, Non-transfusion-dependent thalassemia; PCV, packed cell volume; RBC, red blood cell count; RDW, red cell distribution; SD, standard deviation; TDT, transfusion-dependent thalassemia. Normal range (as reported in IthaPhen [11]): Hb 11.5–16.5 (female), 13–18 (male); RBC 3.8–5.8 (female), 4.5–6.5 (male); PCV 37–47 (female), 40–52 (male); MCV 80–100; MCH 27–32, MCHC 32–36; HbA_2_ 1.8–3; RDW, <15; HbF, <1.5.

**Table 3 biomedicines-12-00296-t003:** Phenotype associated with β +33 C>G [HBB:c.-18C>G] in heterozygous individuals.

Case	Sex-Age	Hb g/dL	RBC ×10^6^	RDW %	PCV %	MCV fL	MCH pg	MCHC g/dL	HbA_2_ %	α Genotype
1	F-66	13.7	4.86		41.5	85.4	28.2	33	2.8	
2	F-30	14	4.79		40.7	85	29.2	32.6	3.2	
3	F-23	11.6	4.53		38	83.9	25.6	30.5	2.8	αα/-α^3.7^
4	F-30	11.9	4.23	12.9	36.9	87.1	28	32.2	3	αα/αα
5	F-65	14.3	4.93	14.1	42.1	85.4	29	34	2.9	
6	F-56	13.9	5.35		42.4	79.3	26	32.8	2.4	αα/-α^3.7^
7	F-23	11.7	4.97	18.7	36.4	73.2	23.5	32.1	2.7	αα/αα
8	F-44	14	4.48	13.3	41.9	93.5	31.3	33.4	2.7	
9	F-51	12.7	4.9	13.9	40	81.6	25.9	31.8	2.6	
10	F-58	13.6	4.43	13.6	40.1	90.5	30.7	33.9	3.3	
11	F-27	13.1	4.56	13.2	39.6	86.8	28.7	33.1	3.3	
12	F-27					69.9	22.7		3.5	-α^3.7^/-α^3.7^
13	F-22					90.3	30.2		3	
14	F-25					91	29.6		3.2	
15	F-16					90.1	28.5		3	
16	F-21	13.2	4.66		39.3	84.3	28.3	33.6	2.9	
17	M-70	13.7	4.98		41.7	83.7	27.5	32.9	2.9	
18	M-33	15.7	5.29	11.9	47.8	90.3	29.7	32.9	3	αα/αα
19	M-34	17.2	6.22		49.9	80.2	27.7	34.7	3.1	
20	M-32	15.5	5.19	13.5	46.3	89.2	29.9	33.5	3.3	
21	M-26					88.2	28.3		3	
22	M-38	14.8	5.35	13.8	43.6	81.5	27.7	33.9	3	
23	M-25					86.9	29		3	
24	M-17					85.3	27.7		3.1	
25	M-22					83.6	27.9		3.2	
26	M-25	15.9	5.27		46.8	88.8	30.2	34	3.6	

F, females; M, males; Hb, hemoglobin; RBC, red blood cell count; RDW, red cell distribution; PCV, packed cell volume; MCV, mean corpuscular volume; MCH, mean corpuscular hemoglobin; MCHC, mean corpuscular hemoglobin concentration; HbA_2_, A_2_ hemoglobin.

**Table 4 biomedicines-12-00296-t004:** Phenotype associated with β +33 C>G [HBB:c.-18C>G] in compound heterozygous individuals.

Case	1	2	3	4	5	6	7	8	9	10	11
Sex/Age	F/46	F/61	F/58	F/23	M/4	M/23	M/24	F/25	F/34	M/51	F/15
Hb g/dL	9	9.5	8	8.7	8		13.7	8.7	8.5	8.7	9.6
RBC × 10^6^	5.24	5.13	4.14	4.38	4.64		7.07	4.56	3.99	4.21	4.92
MCV fL	55.5	62.2	60.9	61.2	54.5		58.1	58.6	65.2	64.8	61.8
MCH pg	17.2	18.5	19.4	19.9	17.2		19.4	19.1	21.3	20.7	19.5
MCHC g/dL	30.9	29.8	31.8	32.5	31.6		33.3	32.6	32.7	31.9	31.6
PCV %	29.1	31.9	25.2	26.8	25.3		41.1	26.7	26	27.3	30.4
HbA_2_ %	5.4	NA	6	5.8	6		6.2	6.4	5.8	5.2	5.8
HbF %	3.1	1.6	1.5	4				NA	8.2	2.2	7.2
β/α ratio		0.31			0.25						
Film	++			++++			+	+++	++++	++	+++
β genotype	CD39/+33	CD39/+33	CD39/+33	CD39/+33	IVSI-110/+33	IVSI-110/+33	IVSI-110/+33	IVSI-110/+33	IVSI-110/+33	IVSI-110/+33	IVSI-1/+33
α genotype	αα/αα	αα/αα	αα/αα	αα/αα	αα/αα	αα/αα	αα/-α^3.7^	αα/αα	αα/αα	αα/αα	αα/αα
Xmn*I*	−/−	−/−	−/−	−/−	−/−	−/−	+/−	−/−	+/−	−/−	−/−
Thalassemiaphenotype	NTDT	NTDT	TDT	TDT	NTDT	TDT	NTDT	NTDT			NTDT
Transfusion frequency	Rare (once post birth, once post surgery)	Never	Regular (began at 31 yr)	Regular (began at 35 yr)	Never	Regular (began at 36 mo)	Never	Rare (post both pregnancies)	Never	Never	Never
Bonedeformities	None	None	Facial	None	None	None	None	None			None
Splenomegaly	No	No	Yes	No	Yes	No	No	No			Barely palpable
Hepatomegaly	No	No	No	No	No		No	No			No
Splenectomy	No	No	No	No	No	No	No	No			No

F, females; M, males; Hb, hemoglobin; RBC, red blood cell count; MCV, mean corpuscular volume; MCH, mean corpuscular hemoglobin; MCHC, mean corpuscular hemoglobin concentration; PCV, packed cell volume; HbA_2_, A_2_ hemoglobin; HbF, F hemoglobin; NTDT, non-transfusion-dependent thalassemia; TDT, transfusion-dependent thalassemia; yr, years old; mo, months old; +33, HBB:c.-18C>G; CD39, HBB:c.118C>T; IVS I-110, HBB:c.93-21G>A; IVS I-1, HBB:c.92+1G>A. Case 3 is a cancer patient. Case 4 is diagnosed at age 23 with transfusion therapy requirements documented at age 35. Case 6 is diagnosed with β-thalassemia major and has undergone lifelong transfusions.

## Data Availability

The data presented in this study are openly available via a web interface at https://www.ithanet.eu/db/ithaphen, accessed on 18 September 2023. All data relevant to the study are included in the article or uploaded as online Appendix A.

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
