# Peer review of "Unravelling the Complexity of the +33 C>G [HBB:c.-18C>G] Variant in Beta Thalassemia"

_biomedicines, 2024, doi:10.3390/biomedicines12020296_

Round 1

Reviewer 1 Report

Comments and Suggestions for Authors

It is a well writing paper regarding the genotype-phenotype findings of a new rare mutation of β-Thalassemia occuring more common in Cyprus. It is always valuable to publish data that provide new information which lead the genetic counseling.

One question: how do you explain the very high hemoglobin level of individual No 19?

Reviewer 2 Report

Comments and Suggestions for Authors

In their manuscript „Unravelling the complexity of the +33 C>G [HBB:c.-18C>G] variant in beta thalassemia“ Coralea Stephanou et al conduct a complex study in which they describe a novel molecular variant that can have an aggravating effect in connection with pre-existing minimal thalassemia.

The manuscript describes very impressively the extensive work that was invested over several years and allows not only a genotypic but also a phenotypic classification.

Although the manuscript is very in-depth, there are some comments that the authors could adopt to improve it.

Major:

Unfortunately, Table 3 does not show a rigorous, complete analysis of all laboratory parameters, which were only comprehensively collected in around 18 of 26 clinical cases. Patient 3 is not assigned age.

Table 4, Patient 4. As Year of detection presumably dates to age of 23, year of onset at age 35 cannot be right. In addition, many explanatory laboratory parameters are missing from Table 4, especially in patient 6, who requires regular transfusions.

The pedigree shown in Figure 1 leaves a lot of room for imprecise interpretation. While the genetic situation of the father and index proband is described as a compound heterozygosity with no disease significance (IVS I-110 G>A variant), the mother also has a severe IVS I-110 G>A variant that is not otherwise described. This description needs further clarification.

Minor

Table 2 needs further layout improvements.

Figure 2 shows the essentials but remains in need of improvement in some areas in order to make the importance of the analysis more clear.

Round 2

Reviewer 2 Report

Comments and Suggestions for Authors

I suggest accepting the manuscript in the present form.